# Diversity Policy Gradient for Sample Efficient Quality-Diversity Optimization

## Abstract

A fascinating aspect of nature lies in its ability to produce a large and diverse collection of organisms that are all high-performing in their niche. By contrast, most AI algorithms focus on finding a single efficient solution to a given problem. Aiming for diversity in addition to performance is a convenient way to deal with the exploration-exploitation trade-off that plays a central role in learning. It also allows for increased robustness when the returned collection contains several working solutions to the considered problem, making it well-suited for real applications such as robotics. Quality-Diversity (QD) methods are evolutionary algorithms designed for this purpose. This paper proposes a novel algorithm, QD-PG, which combines the strength of Policy Gradient algorithms and Quality Diversity approaches to produce a collection of diverse and high-performing neural policies in continuous control environments. The main contribution of this work is the introduction of a Diversity Policy Gradient (DPG) that exploits information at the time-step level to thrive policies towards more diversity in a sample-efficient manner. Specifically, QD-PG selects neural controllers from a MAP-Elites grid and uses two gradient-based mutation operators to improve both quality and diversity, resulting in stable population updates. Our results demonstrate that QD-PG generates collections of diverse solutions that solve challenging exploration and control problems while being two orders of magnitude more sample-efficient than its evolutionary competitors.

## 1   Introduction

Natural evolution has the fascinating ability to produce diverse organisms that are all well adapted to their respective niche. Inspired by this ability to produce a tremendous diversity of living systems, Quality-Diversity (QD) is a new family of optimization algorithms that aims at searching for a collection of both diverse and high-performing solutions (Pugh et al., 2016; Cully & Demiris, 2017). While classic optimization methods focus on finding a single efficient solution, QD optimization aims to cover the range of possible solution types and to return the best solution for each type. This process is sometimes referred to as "illumination" in opposition to optimization, as it reveals (or illuminates) a search space of interest often called the *behavior descriptor space* (Mouret & Clune, 2015).

The principal advantage of QD approaches resides in their intrinsic capacity to deliver a large and diverse set of working alternatives when a single solution fails (Cully et al., 2015). By producing a collection of solutions instead of a unique one, QD algorithms allow to obtain different ways to solve a single problem, leading to greater robustness, which can help to reduce the reality gap when applied to robotics (Koos et al., 2012). Diversity seeking is the core component that allows QD algorithms to generate large collections of diverse solutions. By encouraging the emergence of novel behaviors in the population without focusing on performance alone, diversity seeking algorithms explore regions of the behavior descriptor space that are unreachable for conventional algorithms (Doncieux et al.,

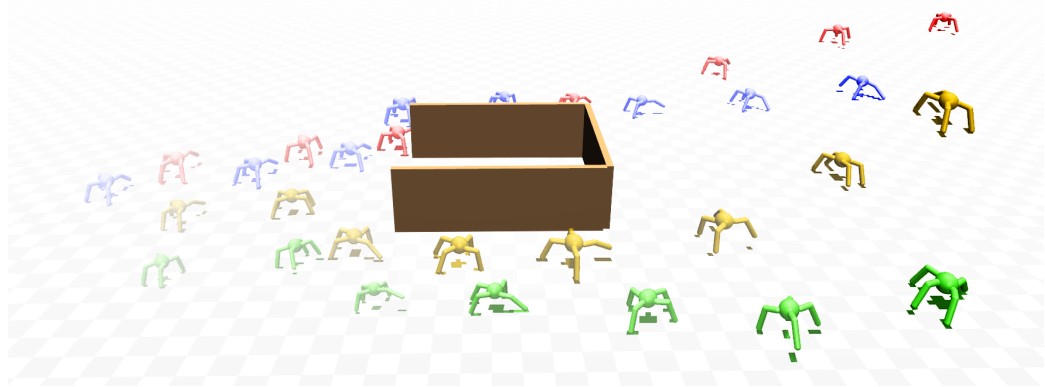

Figure 1: The agent robot is rewarded for running forward as fast as possible. Following the reward signal without further exploration leads the agent into the trap, which corresponds to a poor local minimum. QD-PG produces a collection of solutions that are diverse and high-performing, allowing to find several working alternatives to solve a deceptive control problem.

2019). Another benefit of QD is its ability to solve hard exploration problems where the reward signal is sparse or deceptive, and on which standard optimization techniques are ineffective (Colas et al., 2020). This ability can be interpreted as a direct consequence of the structured search for diversity in the behavior descriptor space.

Quality-Diversity algorithms build on black-box optimization methods such as evolutionary algorithms to evolve a population of solutions (Cully & Demiris, 2017). Historically, they rely on random mutations to explore small search spaces but struggle when facing higher-dimensional problems. As a result, they often scale poorly to problems where neural networks with many parameters provide state-of-the-art results (Colas et al., 2020).

Building large and efficient controllers that work with continuous actions has been a long-standing goal in Artificial Intelligence and in particular in robotics. Deep reinforcement learning (RL), and especially Policy Gradient (PG) methods have proven efficient at training such large controllers (Schulman et al., 2017; Lillicrap et al., 2015; Fujimoto et al., 2018; Haarnoja et al., 2018). One of the keys to this success lies in the fact that PG methods exploit the structure of the objective function when the problem can be formalized as a Markov Decision Process (MDP), leading to substantial gains in sample efficiency. Moreover, they also exploit the analytical structure of the controller when known, which allows the sample complexity of these methods to be independent of parameter space dimensionality (Vemula et al., 2019). In real-world applications, these gains turn out to be critical when interacting with the environment is expensive. PG methods usually rely on simple exploration mechanisms, like adding Gaussian noise (Fujimoto et al., 2018) or maximizing entropy (Haarnoja et al., 2018) to explore the action space, which happens to be insufficient in hard exploration tasks where the reward signal is sparse or deceptive (Colas et al., 2018; Nasiriany et al., 2019).

Successful attempts have been made to combine evolutionary methods and reinforcement learning (Khadka et al., 2019; Khadka & Tumer, 2018; Pourchot & Sigaud, 2018; Shi et al., 2020). However, all these techniques only focus on building high-performing solutions and do not explicitly encourage diversity within the population. In this regard, they fail when confronted with hard exploration problems. To address these problems, one needs to seek both high-performing solutions and diversity within them.

**Contributions**

In this work, we introduce the idea of a *diversity policy gradient* (DPG) that thrives solutions towards more diversity. We show that the DPG can be used in combination with the standard policy gradient, dubbed *quality policy gradient* (QPG), to produce high-performing and diverse solutions. Our algorithm, called QD-PG, builds on MAP-Elites (Mouret & Clune, 2015), demonstrates remarkable sample efficiency brought by off-policy PG methods, and produces collections of good solutions in a single run (see Figure 1). We compare QD-PG to state-of-the-art RL algorithms and to several evolutionary methods known as Evolution Strategies (ESs) augmented with a diversity objective,

73 namely the NS-ES family (Conti et al., 2018) and the ME-ES algorithm (Colas et al., 2020). We
74 show that QD-PG generates collections of robust solutions in hard exploration problems while RL
75 algorithms struggle to produce a single one, and that QD-PG is two orders of magnitude more sample
76 efficient than the best of its evolutionary competitors.

## 2 Background

**Problem statement**

79 We consider an MDP $(\mathcal{S}, \mathcal{A}, \mathcal{R}, \mathcal{T}, \gamma)$ where $\mathcal{S}$ is the state space, $\mathcal{A}$ the action space, $\mathcal{R} : \mathcal{S} \times \mathcal{A} \to \mathbb{R}$
80 the reward function, $\mathcal{T} : \mathcal{S} \times \mathcal{A} \to \mathcal{S}$ the dynamics transition function and $\gamma$ a discount factor.
81 We assume that both $\mathcal{S}$ and $\mathcal{A}$ are continuous and consider a controller, or policy, $\pi_\theta : \mathcal{S} \to \mathcal{A}$
82 parameterized by $\theta \in \Theta$, which is called a *solution* to the problem. We say that a solution $\theta$ is
83 *highy-performing* if the expectation over the sum of rewards is high when using $\pi_\theta$. The *fitness* of a
84 solution measures its performance $F : \Theta \to \mathbb{R}$ where $F(\theta) = \mathbb{E}_{\pi_\theta} \sum_t \gamma^t r_t$.

85 To characterize the novelty of a solution w.r.t. $J$ other solutions, as in QD methods, we introduce a
86 behavior descriptor (BD) space $\mathcal{B}$, a behavior descriptor extraction function $\xi : \Theta \to \mathcal{B}$, and define a
87 distance metric $||.||_{\mathcal{B}}$ over $\mathcal{B}$. The *novelty* $n : \Theta \times \Theta^J \to \mathbb{R}^+$ of a solution $\theta$ w.r.t. a list of solutions
88 $(\theta_j)_{j=1,...,J}$ is defined as $n\left(\theta, (\theta_j)_{j=1,...,J}\right) = \sum_j ||\xi(\theta), \xi(\theta_j)||_{\mathcal{B}}$. In other words, we quantify the
89 novelty of a solution w.r.t. a list of $J$ solutions as the sum of distances between its behavior descriptor
90 and the behavior descriptors of all solutions of the list. We also use the distance $||.||_{\mathcal{B}}$ to characterize
91 the *diversity* of a set of $K$ solutions $\{\theta_k\}_{k=1,...,K}$. We formally define diversity $d : \Theta^K \to \mathbb{R}^+$ as

$$d\left(\{\theta_k\}_{k=1,...,K}\right) = \sum_{i=1}^{K} \min_{k \neq i} ||\xi(\theta_i), \xi(\theta_k)||_{\mathcal{B}}, \tag{1}$$

92 meaning that a set of solutions is diverse if the solutions are distant with respect to each other in the
93 sense of $||.||_{\mathcal{B}}$.

**The MAP-Elites algorithm**

95 MAP-Elites (Mouret & Clune, 2015) is a simple yet state-of-the-art QD algorithm that has been
96 successfully applied to a wide range of challenging problems such as robot damage recovery (Cully
97 et al., 2015), molecular robotic control (Cazenille et al., 2019) and game design (Alvarez et al., 2019).
98 In MAP-Elites, the behavior descriptor space $\mathcal{B}$ is discretized into a grid of cells, also called niches,
99 with the aim of filling each cell with a high-performing solution. The algorithm starts with an empty
100 grid and an initial random set of $K$ solutions that are evaluated and added to the grid by following
101 simple insertion rules. If the cell corresponding to the behavior descriptors of a solution is empty, then
102 the solution is added to this cell. If there is already a solution in the cell, the new solution replaces it
103 only if it has greater fitness. At each iteration, $P$ existing solutions are sampled uniformly from the
104 grid and randomly mutated to create $P$ new solutions. These new solutions are then evaluated and
105 added to the grid following the same insertion rules. This cycle is repeated until convergence or for a
106 given budget of iterations.

107 Though MAP-Elites is a compelling and efficient method, it suffers from a low sample efficiency
108 as it relies on random mutations. Recently, Colas et al. (2020) tackled this problem by updating
109 the solutions through an Evolution Strategy known as the Cross-Entropy method. Notably, they
110 showed that MAP-Elites could be scaled with their method to address complex MuJoCo control
111 environments at the cost of very large computational resources. In this study, we propose to harness
112 policy gradients (QPG and DPG) to build a more sample-efficient MAP-Elites approach.

## 3 Key Principle: Diversity Policy Gradient

114 Let us assume that we have a MAP-Elites grid containing $K$ solutions $(\theta_1, \ldots, \theta_K)$. To increase
115 diversity in the grid using the DPG, we need to update one sampled solution $\theta$ from the grid using
116 gradient ascent. To do so, we aim to compute the gradient of the population diversity w.r.t. $\theta$, where
117 diversity is defined in Equation (1). As the $K$ solutions are independent, order does not matter and

we can consider optimizing arbitrarily $\theta = \theta_1$. To compute the gradient of $d$ w.r.t. $\theta_1$, we need to separate the terms that depend on $\theta_1$ from the others. The terms that depend on $\theta_1$ correspond to the distance of $\theta_1$ to its nearest neighbor, which we define as $\theta_2$, and to the distances of $\theta_1$ to the $\theta$s for which $\theta_1$ is the nearest neighbor. We can arbitrarily index them from 3 to $J$[1], thus:

$$d(\{\theta_k\}_{k=1,...,K}) = \sum_{j=2}^{J} ||\xi(\theta_1), \xi(\theta_j)||_{\mathcal{B}} + M, \text{ where } M = \sum_{i \notin \{1,...,J\}} \min_{k \neq i} ||\xi(\theta_i), \xi(\theta_k)||_{\mathcal{B}}.$$

Only the first term of the sum depends on $\theta = \theta_1$. Furthermore, we observe that this term equals the novelty of solution $\theta_1$ w.r.t. the list $(\theta_j)_{2 \leq j \leq J}$. Therefore, the gradient of diversity w.r.t. $\theta_1$ is

$\nabla_{\theta_1} d(\{\theta_k\}_{k=1,...,K}) = \nabla_{\theta_1} n(\theta_1, (\theta_j)_{2 \leq j \leq J})$. That is, we can increase the diversity of the population by increasing the novelty of $\theta_1$ w.r.t. the list $(\theta_j)_{2 \leq j \leq J}$. In practice, we replace this list by a list of nearest neighbors of $\theta_1$, as this is easier to compute and the elements of $(\theta_j)_{2 \leq j \leq J}$ tend to be among the nearest neighbors of $\theta_1$.

Under this form, the diversity gradient cannot benefit from the variance reduction methods in the RL literature to efficiently compute policy gradients Sutton et al. (1999). To this end, we need to express it as a gradient over the expectation of a sum of scalar quantities obtained by policy $\pi_{\theta_1}$ at each step when interacting with the environment. Therefore, to build a DPG, we need information about the novelty of a solution at the time step level. To do so, we introduce a novel space $\mathcal{D}$, dubbed *state descriptor space* and a *state descriptor extraction function* $\psi : \mathcal{S} \rightarrow \mathcal{D}$. We assume $\mathcal{D}$ and $\mathcal{B}$ have the same dimension. Similarly to the novelty of a solution, we now define the novelty of a state $s$ w.r.t. $J$ other states $(s_j)_{j=1,...,J}$ as $n : \mathcal{S} \times \mathcal{S}^J \rightarrow \mathbb{R}$ such that $n(s, (s_j)_{j=1,...,J}) = \sum_{j=1}^{J} ||\psi(s), \psi(s_j)||_{\mathcal{D}}$, where $||.||_{\mathcal{D}}$ is a distance metric over $\mathcal{D}$.

Now, we need to link novelty defined at the time step level to novelty defined at the solution level. We define the novelty of a state w.r.t. a set of solutions. We say that a state is novel w.r.t. some solutions if the state is novel w.r.t. to the states visited by these solutions. More formally:

$$n(s, (\theta_j)_{j=1,...,J}) = \sum_{j=1}^{J} \mathbb{E}_{\pi_{\theta_j}} \sum_{t} ||\psi(s), \psi(s_t)||_{\mathcal{D}}. \tag{2}$$

While we adopt this definition in this paper, one might as well consider other definitions where, for instance, a state is compared to states that have been visited at the same time step during another episode. In this context, if the following relation is satisfied:

$$\mathbb{E}_{\pi_{\theta_1}} \sum_{t} n(s_t, (\theta_j)_{2 \leq j \leq J}) = n(\theta_1, (\theta_j)_{2 \leq j \leq J}), \tag{3}$$

then we can compute the DPG of $d$ w.r.t. $\theta_1$ as

$$\nabla_{\theta_1}^{DPG} = \nabla_{\theta_1} \mathbb{E}_{\pi_{\theta_1}} \sum_{t} n(s_t, (\theta_j)_{2 \leq j \leq J}). \tag{4}$$

This expression corresponds to the classical policy gradient setting where $\gamma = 1$ and where the corresponding reward signal, here dubbed diversity reward, is computed as $r_t^D = n(s_t, (\theta_j)_{2 \leq j \leq J})$. Therefore, this gradient can be computed using any PG estimation technique replacing the environment reward by the diversity reward $r_t^D$.

Equation (3) enforces a relation between $\mathcal{B}$ and $\mathcal{D}$ and between extraction functions $\psi$ and $\xi$. In practice, it may be hard to define the behavior descriptor and state descriptor of a solution that satisfy this relation while being meaningful to the problem at hand and tractable. But a strict equality is not necessary. It suffices that an increase on the left-hand side implies an increase on the right-hand side so that we can still update $\theta_1$ using (4). Furthermore, when this is not the case, the diversity gradient update might not result in an increase of diversity in the behavior descriptor space, but in that case the MAP-Elites insertion rule will remove the corresponding solution. We show in Section 6 that we can define descriptors that do not satisfy the above relation all the time, but still give satisfactory results.

---

[1]Remark: $\theta_2$ can appear twice in the list $(\theta_j)_{2 \leq j \leq J}$

## 4    Related Work

A distinguishing feature of our approach is that we combine diversity seeking at the level of trajectories using behavior descriptors and diversity seeking in the state space using state descriptors. The former is used by MAP-Elites to select solutions from the grid and contributes structural bias towards diversity, whereas the latter is used during policy gradient steps in the RL part, see Figure 2b. We organize the literature review below according to this split between two types of diversity seeking mechanisms.

**QD search in the solution space**

Simultaneously maximizing diversity and performance is the central goal of QD methods (Pugh et al., 2016; Cully & Demiris, 2017). Among the various possible combinations offered by the QD framework, Novelty Search with Local Competition (NSLC) (Lehman & Stanley, 2011b) and MAP-Elites (Mouret & Clune, 2015) are the two most popular algorithms. NSLC builds on the Novelty Search (NS) algorithm (Lehman & Stanley, 2011a) and maintains an unstructured archive of solutions selected for their local performance while MAP-Elites uniformly samples individuals from a structured grid that discretizes the BD space. Not clear in its current form. I suggest: "QD-PG uses the standard grid of MAP-Elites. However, we also show in Appendix F that QD-PG can be used with alternative archive structures.

With the objective of improving their data-efficiency, QD-ES algorithms that combine QD and ESs, such as NSR-ES and NSRA-ES, have been applied to challenging continuous control environments in Conti et al. (2018). But, as outlined in Colas et al. (2020), they suffer from poor sample efficiency and the diversity and environment reward functions could be mixed in a more efficient way. In that respect, the most closely related work w.r.t. ours is ME-ES (Colas et al., 2020). The ME-ES algorithm also optimizes quality and diversity using MAP-Elites and two ES populations. Using these methods was shown to be critically more efficient than population-based GA algorithms (Salimans et al., 2017), but our results show that they are still less sample efficient than off-policy deep RL methods, as they do not leverage the analytical computation of the policy gradient at the time step level. To the best of our knowledge, no QD or ES algorithm use an explicit critic for both performance and diversity, resulting in even higher data-efficiency.

**QD search in the state or action spaces**

Seeking for diversity in the space of states or actions is generally framed into the RL framework. This is the case of algorithms maintaining a population of RL agents for exploration without an explicit diversity criterion (Jaderberg et al., 2017) or algorithms explicitly looking for diversity but in the action space rather than in the state space like ARAC (Doan et al., 2019), P3S-TD3 (Jung et al., 2020) and DvD (Parker-Holder et al., 2020).

An exception is Stanton & Clune (2016) who define a notion of *intra-life novelty* that is similar to our state novelty defined in Section 3. However, their novelty relies on skills rather than states. Our work is also related to algorithms using RL mechanisms to search for diversity only (Eysenbach et al., 2018; Pong et al., 2019; Lee et al., 2019; Islam et al., 2019). These methods have proven useful in sparse reward situations, but they are inherently limited when the reward signal can orient exploration, as they ignore it. Other works sequentially combine diversity seeking and RL. The GEP-PG algorithm Colas et al. (2018) combines a diversity seeking component, namely *Goal Exploration Processes* (Forestier et al., 2017) and the DDPG deep RL algorithm (Lillicrap et al., 2015). This sequential combination of exploration-then-exploitation is also present in GO-EXPLORE (Ecoffet et al., 2019). Again, this approach is limited when the reward signal can help driving the exploration process to efficient solutions. These sequential approaches first look for diversity in the behavior descriptor space, then optimize performance in the state action space, whereas we do so simultaneously in the behavior descriptor space and in the state space.

To the best of our knowledge, QD-PG is the first algorithm optimizing both diversity and performance in the solution and in the state space, using a sample-efficient policy gradient computation method for the latter.

## 5    Methods

Our full algorithm is called QD-PG, its pseudo code is given in Appendix A and its architecture is depicted in Figure 2. QD-PG is an iterative algorithm based on MAP-Elites that replaces random

mutations with policy gradient updates. As we consider a continuous action space and want to improve sample efficiency by using an off-policy policy gradient method, we rely on the Twin Delayed Deterministic Policy Gradient (TD3) algorithm (Fujimoto et al., 2018). See Appendix B for a detailed description of TD3.

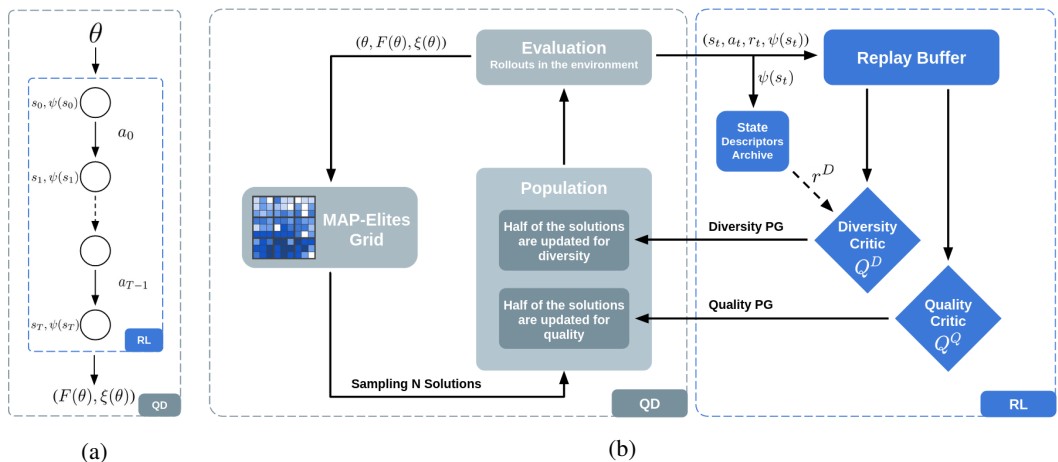

Figure 2: (a): The RL part of QD-PG operates at the time step level while the QD part operates at the controller level, considering the MDP as a black box. (b) One QD-PG iteration consists of three phases: 1) A new population of solutions is sampled from the MAP-Elites grid. 2) These solutions are updated by an off-policy RL agent: half of the solutions are optimized for quality and the other half for diversity. The RL agent leverages one shared critic for each objective. 3) The newly obtained solutions are evaluated in the environment. Transitions are stored in a replay buffer while the updated solutions, their final scores and behavior descriptors are stored in the MAP-Elites grid.

QD-PG maintains three permanent structures. In the QD part, a MAP-Elites grid stores the most novel and performing solutions. In the RL part, a replay buffer contains all transitions collected when evaluating solutions and an archive $\mathbb{A}$ stores all state descriptors obtained so far. QD-PG starts with an initial population of random solutions, evaluates them and inserts them into the MAP-Elites grid. At each iteration, solutions are sampled from the grid, copied, and updated. The updated solutions are then evaluated through one rollout in the environment and inserted into the grid according to insertion rules. Transitions collected during evaluation are stored in the replay buffer, and state descriptors are stored in the archive $\mathbb{A}$. Note that these state descriptors are first filtered to avoid insertion in the archive of multiple state descriptors that are too close to each other.

During the update step, half the population is updated with QPG ascent and the other half with DPG ascent. The choice of whether an agent is updated for quality or diversity is random, meaning that it can be updated for quality and later for diversity if selected again. To justify this design, we show in Section 6 that updating consecutively for quality and diversity outperforms updating based on joint criteria. Both gradients are computed from batches of transitions sampled from the replay buffer. The QPG is computed as usual from rewards whereas for DPG, we get fresh novelty rewards as

$$r_t^D = \sum_{j=1}^{J} ||\psi(s_t), \psi(s_j)||_\mathcal{D}, \tag{5}$$

where $(s_j)_{j=1,\dots,J}$ are the $J$ nearest neighbors of state $s_t$ in the archive $\mathbb{A}$. Diversity rewards must be recomputed at each update because $\mathbb{A}$ changes during training. Following Equation (2), diversity rewards should be computed as the sum of the distances between the descriptor of $s_t$ and the descriptors of all the states visited by a list of $J$ solutions. In practice, we consider the $J$ nearest neighbors of $s_t$. This choice simplifies the algorithm and is faster and works well in practice.

TD3 relies on a parameterized critic to reduce the variance of its policy gradient estimate. In QD-PG, we maintain two parameterized critics $Q_w^D$ and $Q_v^Q$, respectively dubbed diversity and quality critics, every time a policy gradient is computed, QD-PG also updates the corresponding critic. In fact, as in TD3, we use pairs of critics and target critics to fight the overestimation bias. We share the critic parameters among the population as in Pourchot & Sigaud (2018). Reasons for doing so come from

the fact that diversity is not stationary, as it depends on the current population. If each agent had its own diversity critic, since an agent may not be selected for a large number of generations before being selected again, its critic would convey an outdated picture of the evolving diversity. We tried this solution, and it failed. A side benefit of critic sharing is that both critics become accurate faster as they combine experience from all agents. Additional details on QD-PG implementation are available in Appendix C.

## 6 Experiments

In this section, we intend to answer the following matters: 1. Can QD-PG produce collections of diverse and high-performing neural policies and what are the advantages to do so? 2. Is QD-PG more sample efficient than its QD competitors? 3. To what extent are the considered benchmarks difficult for classical policy gradients methods? 4. What is the usefulness of the different components of QD-PG?

**Environments**

We asses QD-PG capabilities in continuous control environments that exhibit high dimensional observation and action spaces as well as strong exploration difficulties. Two types of reward signals, dubbed sparse and deceptive, are known to be particularly difficult for classical RL methods. These rewards appear in many applications such as robotics or combinatorial optimization. Sparse rewards are obtained if a given condition is specified, leading to a majority of null rewards and to credit assignment difficulties. Deceptive rewards are dense signals, i.e., they are non-zero at each time step but can mislead the search process to some local optimum. In such problems, a good approach to the exploration-exploitation trade-off is essential. The agent should learn when to ignore the reward signal and explore to avoid local minima and when to follow it to increase its return. Deceptive environments constitute a natural choice to highlight QD efficiency to balance exploration and exploitation. In this study, we consider three OpenAI Gym environments based on the MuJoCo physics engine that all exhibit strong deceptive rewards (illustrated in Appendix 5). Such environments have been widely used in previous works (Parker-Holder et al., 2020; Colas et al., 2020; Frans et al., 2018; Shi et al., 2020) for their deceptive nature, a characteristic that is absent of more widespread continuous control environments like HALFCHEETAH-V2, HOPPER-V2 or still ANT-V2.

In the POINT-MAZE environment, an agent represented as a green sphere must find the exit of the maze depicted in Figure 4a, represented as a red sphere. An observation contains the agent position at time $t$, and an action corresponds to position increments along the $x$ and $y$ axes. The reward is expressed as the negative Euclidean distance between the center of gravity of the agent and the exit center. The trajectory length cannot exceed 200 steps.

The ANT-MAZE environment is modified from OpenAI Gym ANT-V2 (Brockman et al., 2016) and also used in (Colas et al., 2020; Frans et al., 2018). In ANT-MAZE, a four-legged ant has to reach a goal zone located in the lower right part of the maze (colored in green in Figure 4b). Its initial position is sampled in a small circle located in the maze's extreme bottom left. As in POINT-MAZE, the reward is expressed as the negative Euclidean distance between the ant and the center of the goal zone. Maze walls are organized so that following the gradient of the reward function drives the ant into a dead-end. In ANT-MAZE, the final performance is defined as the maximum reward received during an episode. The environment is considered solved when an agent obtains a score superior to $-10$, corresponding to reaching the goal zone. An episode consists of 3000 time steps, this horizon is three times larger than in usual MuJoCo environments, making this environment particularly challenging for RL based methods (Vemula et al., 2019).

Finally, the ANT-TRAP environment also derives from ANT-V2 and is inspired from (Colas et al., 2020; Parker-Holder et al., 2020). In ANT-TRAP, the four-legged ant initially appears in front of a trap and must bypass it to run as fast as possible in the forward direction (see Figure 4c), as in ANT-V2, the reward is computed as the ant velocity on the x-axis. The trap consists of three walls forming a dead-end directly in front of the ant, leading to a strong deceptive reward. In this environment, the trajectory length cannot exceed 1000 steps. As opposed to POINT-MAZE and ANT-MAZE, where the objective is to reach the exit area, there is no unique way to solve ANT-TRAP and we expect a QD algorithm to generate various effective solutions as depicted in Figure 1.

**Baselines and Ablations**

QD-PG is compared to three types of methods. First, to answer question 2, we compare QD-PG to a family of QD baselines, namely ME-ES, NSR-ES, and NSRA-ES (Colas et al., 2020). Appendix E.1 recaps the properties of all these methods. Second, to answer question 3, we compare QD-PG to a family of policy gradient baselines. Soft Actor Critic (SAC) (Haarnoja et al., 2018) and the Twin Delayed Deep Deterministic policy gradient (TD3) (Fujimoto et al., 2018) are continuous control algorithms achieving state-of-the-art results on MUJOCO benchmarks. Random Network Distillation (RND) (Burda et al., 2018) is a curiosity-driven RL agent (Schulman et al., 2017) which was shown to perform well in hard exploration settings. CEM-RL (Pourchot & Sigaud, 2018) mixes Cross-Entropy Methods (CEM) and RL to evolve a population of agents to maximize quality and obtains state-of-the-art results MUJOCO benchmarks. Finally, to answer question 4, we propose to investigate the following matters: Can we replace alternating quality and diversity updates by a single update that optimizes for the sum of both criteria? Are quality gradients updates alone enough to fill the MAP-Elites grid? Are diversity gradients updates alone enough to do so? Consequently, we consider the following ablations of QD-PG: QD-PG SUM computes a gradient to optimize the sum of the quality and diversity rewards, D-PG applies only diversity gradients to the solutions, and Q-PG applies only quality gradients, but both D-PG and Q-PG still use QD selection (see Appendix E.1).

We compare QD-PG to its ablations and RL competitors in all environments and show results in Table 1a. Detailed results including graphic charts and coverage maps are given in Appendix E and more details about the evaluation procedure are given in Appendix E.1.

# 7 Results

**1. Can QD-PG produce collections of neural policies and what are the advantages to do so?**

Table 1a presents QD-PG performances. In all environments, our algorithm manages to find working solutions that avoid local minima and reach the overall objective. In addition to its exploration capabilities, QD-PG generates collections of high performing solutions in a single run. During the ANT-TRAP experiment, the final collection of solutions returned by QD-PG contained, among others, 5 solutions that were within a 10% performance margin from the best one. As illustrated in Figure 1, these agents typically differ in their gaits and preferred trajectories to circumvent the trap.

Generating a collection of diverse solutions comes with the benefit of having a repertoire of diverse solutions that can be used as alternatives when the MDP changes (Cully et al., 2015). We show that QD-PG is more robust than conventional policy gradient methods by changing the reward signal of the ANT-MAZE environment. We replace the original goal in the bottom right part of the maze (see Figure 3) with a new randomly located goal in the maze. Instead of running QD-PG to optimize for this new objective, we run a Bayesian optimization process to quickly find a good solution among the ones already stored in the grid. With a budget of only 20 solutions to be tested during the Bayesian optimization process, we are able to quickly recover a good solution for the new objective. We repeat this experiment 100 times, each time with a different random goal, and obtain an average performance of $-10$ with a standard deviation of 9. In other words, 20 interaction episodes (corresponding to 60.000 time steps) suffice for the adaptation process to find a solution that performs well for the new objective without the need to re-train agents. More detailed results can be found in Appendix E.3. [2]

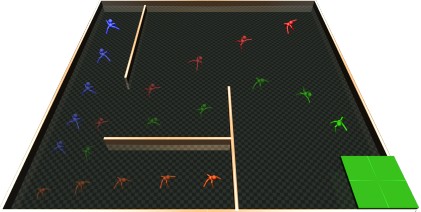

Figure 3: QD-PG produces a collection of diverse solutions. In ANT-MAZE, even after setting new randomly located goals, the MAP-Elites grid still contains solutions that are suited for the new objectives.

**2. Is it more sample efficient than its QD competitors?**

Table 1b compares QD-PG to Deep Neuroevolution algorithms with a diversity seeking component in terms of sample efficiency. QD-PG runs on 10 CPU cores for 2 days while its competitors used 1000 CPU cores for the same duration. Nonetheless, QD-PG matches the asymptotic performance of ME-ES using two orders of magnitude fewer samples, explaining the lower resource requirements.

---

[2]Videos of QD-PG agents are available at: `https://sites.google.com/view/qd-pg`

Table 1: Results for all environments. **Final Perf.** is the minimum distance to the goal in ANT-MAZE and the episode return in POINT-MAZE and ANT-TRAP. The **Ratio to ours** column compares the sample efficiency of a method to QD-PG.

(a) Comparison to ablations and PG baselines.

| Algorithm | **Final Perf.** (± std) | | |
|---|---|---|---|
| | POINT-MAZE | ANT-MAZE | ANT-TRAP |
| QD-PG | −**24**(±**0**) | −7(±7) | **1541**(±**86**) |
| QD-PG SUM | −25(±1) | −5(±3) | 1018(±6) |
| D-PG | −37(±3) | −**2**(±**0**) | 1016(±8) |
| Q-PG | −128(±0) | −26(±0) | 1175(±79) |
| CEM-RL | −312(±1) | −26(±0) | 934(±22) |
| SAC | −127(±1) | −59(±1) | 1049(±21) |
| TD3 | −130(±2) | −26(±0) | 1131(±7) |
| RND | −35(±10) | −27(±1) | 978(±61) |

(b) Comparison to evolutionary competitors.

| Algorithm | ANT-MAZE | | |
|---|---|---|---|
| | Final Perf. | Steps to goal | Ratio to ours |
| QD-PG | −7(±7) | **1.15e8** | **1** |
| CEM-RL | −26(±0) | ∞ | ∞ |
| ME-ES | −5(±1) | 2.4e10 | 209 |
| NSR-ES | −26(±0) | ∞ | ∞ |
| NSRA-ES | −**2**(±1) | 2.1e10 | 182 |

We see three reasons for the improved sample efficiency of QD-PG: 1) QD-PG leverages a replay buffer and can re-use each sample several times. 2) QD-PG leverages novelty at the state level and can exploit all collected transitions to maximize quality and diversity. For instance, in ANT-MAZE, a trajectory brings 3000 samples to QD-PG while standard QD methods would consider it a unique sample. 3) PG exploits the analytical gradient between the neural network weights and the resulting policy action distribution and estimates only the impact of the distribution on the return. By contrast, standard QD methods directly estimate the impact on the return of randomly modifying the weights.

**3. To what extent the considered benchmarks are difficult for policy gradients methods?**

Table 1a compares QD-PG to state-of-the-art policy gradient algorithms and validates that classical policy gradient methods fail to find optimal solutions in deceptive environments. TD3 quickly converges to local minima of performance resulting from being attracted in dead-ends by the deceptive gradients. While we may expect SAC to better explore due to entropy regularization, it also converges to that same local minima in ANT-TRAP and POINT-MAZE. Besides, despite its exploration mechanism based on CEM, CEM-RL also quickly converges to local optima in all benchmarks, confirming the need for a dedicated diversity seeking component. RND, which adds an exploration bonus used as an intrinsic reward (see Appendix G for more details), also demonstrates performances inferior to QD-PG in all environments but manages to solve POINT-MAZE. In ANT-MAZE and ANT-TRAP, as shown in Appendix G.2, RND extensively explores the BD space but fails to obtain high returns.

**4. What is the usefulness of the different components of QD-PG ?**

The ablation study in Table 1a shows that when maximising quality only, Q-PG fails due to the deceptive nature of the reward and when maximizing diversity only, D-PG sufficiently explores to solve the problem in both POINT-MAZE and ANT-MAZE but requires more steps and finds lower-performing solutions. When optimizing simultaneously for quality and diversity, QD-PG SUM fails to learn in ANT-TRAP and manages to solve the task in ANT-MAZE but requires more samples than QD-PG. We hypothesize that quality and diversity rewards may give rise to conflicting gradients. For instance, at the beginning of training in ANT-TRAP, the quality reward drives the ant forward whereas the diversity reward drives it back to escape the trap and explore the environment. Therefore, both rewards cancel each other, preventing any learning. This study validates the usefulness of QD-PG components: 1) optimizing for diversity is required to overcome the deceptive nature of the reward; 2) adding quality optimization provides better asymptotic performance; 3) it is better to disentangle quality and diversity updates.

# 8 Conclusion

This paper is the first to introduce a diversity gradient to explore diversity both at the state and skill levels. Based on this component we proposed a novel algorithm, QD-PG, inspired from the Quality-Diversity literature, that produces collections of diverse and high-performing neural policies in a sample-efficient manner. We showed experimentally that QD-PG generates several solutions that achieve high returns in challenging exploration problems. Finally, we demonstrated that in a few interactions with the environment, QD-PG finds alternative solutions that still obtain good performance when the MDP changes.

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
