# OpenReview forum: "Diversity Policy Gradient for Sample Efficient Quality-Diversity Optimization"
_NeurIPS.cc/2021/Conference — NeurIPS 2021 Submitted_

### Official Review · Reviewer_TBUP · 2021-07-13

**Rating:** 6
**Confidence:** 4

**Summary:**

This paper proposes a Quality-Diversity and Policy Gradient (QD-PG) algorithm that deals with exploration-exploitation trade-off in the optimization problem. The authors validate this algorithm through several benchmark tasks in RL.  The results demonstrate that QD-PG produces competitive solutions especially in the deceptive tasks. The motivation of this paper is clear and the comparison experiments are conducted extensively.

**Ethical Concerns:**

No.

**Limitations And Societal Impact:**

Yes.

**Main Review:**

Although the idea of combining diversity and quality is not novel, the way of combining QD and PG in this paper is interesting. The cost is the extra evaluation plus the definition of a good metric for measuring the diversity. Some comments need to be clarified:
1. How are the two critics (diversity and quality) updated? In the paper, it is not clearly stated. I believe that the update of the ciritic plays an important role for the performance of this algorithm.
2. Is QD-PG more sample efficient than its QD competitors? For the explanation of this research question, can the authors give some quantative results on the TD3 as well?
3. How many replicates of experiments in Table 1?
4. Can you explain why QD-PG SUM fails to learn the ANT-TRAP?

**Time Spent Reviewing:**

5

---

> ### Author Response · Authors · 2021-08-10
> **Response to Reviewer TBUP**
>
> We warmly thank reviewer TBUP for her/his review and very pertinent questions. Here are our answers:
>
> - **1. How are the two critics (diversity and quality) updated? In the paper, it is not clearly stated. I believe that the update of the critic plays an important role for the performance of this algorithm.**
>
> QDPG maintains two parameterized critics that are shared among the population: the quality critic and the diversity critic. As stated in line 233 (or in the pseudo-code in Appendix A), every time a policy gradient (either QPG or DPG) is computed to update a neural policy, QDPG also updates the corresponding critic. We rely on the TD3 agent and give details about the critics update in Appendix B. The main difference between quality and diversity updates lies in the reward used. For the quality update, we simply use the environment reward that is stored in each transition while for diversity update, we first compute fresh diversity rewards when sampling a batch in the replay buffer, as shown in Equation 5.
>
> - **2. Is QD-PG more sample efficient than its QD competitors? For the explanation of this research question, can the authors give some quantitative results on the TD3 as well?**
>
> It is hard to fairly compare QDPG to TD3 in terms of sample efficiency as TD3 falls into the deceptive trap in all environments and thus fails to reach equivalent asymptotic performance. However, if we compare the number of samples required for QDPG to reach the maximum performance found by TD3, then we observe that QDPG has similar sample efficiency. This comparison and other details are in the training curves in Appendix E.1. We observe that same tendency with SAC.
>
> - **3. How many replicates of experiments in Table 1?**
>
> We used 5 randomly selected seeds for each method in each environment. We will add this information in the final version.
>
> - **4. Can you explain why QD-PG SUM fails to learn the ANT-TRAP?**
>
> We assume that QD-PG SUM failing to learn in AntTrap comes from the fact that quality and diversity rewards can cancel each other out in many situations, thus canceling the learning signal. For instance, imagine a situation where the space in front of the Ant has been previously explored. In such a situation, the diversity reward would push the Ant backward whereas the quality reward would push it forward, thus annealing themselves. In this situation, we find it more convenient to apply gradients separately rather than creating a single gradient from the sum of both rewards.

---

### Official Review · Reviewer_76VX · 2021-07-16

**Rating:** 7
**Confidence:** 2

**Summary:**

This paper introduces Diversity Policy Gradient (DPG), an algorithm designed to discover diverse neural policies to solve RL problems in a sample efficient manner. The authors adopt a MAP-Elites scheme to select and evolve neural policies however with the additional twist of updating the policies not just by their quality but also according to their novelty from other known solutions. They show their results on several RL challenges where their algorithm successfully avoids local minima and outperforms other known algorithms such as TD3.

**Limitations And Societal Impact:**

Societal impact is limited in it's current state, though I find that the approach *could* have useful implications for enabling different policies that while equally performant, may have beneficial secondary aspects like ethical superiority.

The main limitation of the study in my view is that doesn't help prescribe how to pick among multiple neural policies, and does not investigate a comparison between found solutions, i.e. in what sense is "diversity" in solutions more useful than just increasing the chance of finding very good ones.

**Main Review:**

While I'm somewhat familiar with QD algorithms, I'm not a domain expert, therefore I'm unable to judge the significance/originality of this paper. I found the idea of using QD to evolve neural policies interesting and to me it was novel.

I found the paper was generally clearly written, and in fact I appreciated the latter parts of the paper where the authors guided the reader to some relevant questions that the paper aims to address. Overall, I found the paper methodologically sound (e.g. ablation studies, benchmarks, and tasks all seemed reasonable and well-motivated), and the communication was quite clear.

From my perspective this is a well-executed paper, of interest to NeurIPS community.

Minor comment:
- There seems to be a leftover comment in the text in line 170 ("Not clear in it's current form...")

----
Questions:

It was not clear to me how the behavior descriptor and state descriptor extraction functions behave and if the choice of these functions were impactful in whether the algorithm works properly. I would appreciate further clarification in the paper. Are these affine or euclidean maps? Does it matter?

I'm also wondering if these mappings preserve some notion of proximity of solutions, and in some sense allow for "transfer" of good trajectory level ideas between policies implicitly through the grid-level mutation-selection balance? If not, do the authors have ideas about whether this information can be exploited?

I wonder if there is a better update schedule than random diversity and quality updates. The authors mention joint updates performed worse. It's not necessary, but I suggest I authors experiment with annealing/oscillating approaches to DPG/QPG ratios for the population, with a larger period than a single update. My main concern is that perhaps QPG ascent benefits from consecutive updates to find better solutions faster, and the current approach "destroys" the gains from the gradient, leading to sample inefficiency.

**Time Spent Reviewing:**

5

---

> ### Author Response · Authors · 2021-08-10
> **Response to Reviewer 76VX**
>
> We warmly thank reviewer 76VX for her/his review and the pertinent questions. Our answers are the following:
>
> - **It was not clear to me how the behavior descriptor and state descriptor extraction functions behave and if the choice of these functions were impactful in whether the algorithm works properly. I would appreciate further clarification in the paper. Are these affine or euclidean maps? Does it matter?**
>
> Indeed, in QD methods, the choice of the behavior descriptor matters and can affect the performance of the algorithm. However, as stated in several references, this choice is always easy to do and can even be considered as an extra degree of freedom given to the user. Regarding the choice of the state extractor descriptor, it is a choice similar to the one for the goal extractor in goal-conditioned methods such as Hindsight Experience Replay and we also found this choice easy to make in practice as all choices that seemed relevant worked. Most of the time, in QD methods, one tries to align the behavior descriptor with the task so that any diversity improvement can also benefit quality and vice-versa. However, it is intractable in some cases to do so. That is why we introduced the Ant-Trap environment in which the descriptors are related to position while performance is related to velocity. Despite this extra difficulty, we observe that the algorithm performs well. We will clarify this part upon acceptance. Regarding the mapping of the behavior descriptor space, we currently use simple Euclidean mapping as it works well in our applications (we will more clearly mention this in the final version) but other mappings could work and we aim to investigate them in future work.
>
> - **I'm also wondering if these mappings preserve some notion of proximity of solutions, and in some sense allow for "transfer" of good trajectory level ideas between policies implicitly through the grid-level mutation-selection balance? If not, do the authors have ideas about whether this information can be exploited?**
>
> In the standard case, the parameter space of policies is high-dimensional, while behavior descriptors (BD) are low-dimensional. As a result, a large variety of policies yield identical behavior descriptors, which means that identical BD mappings do not imply proximity at the trajectory level. However, it has been noted [1, 2] that algorithms which optimize for quality eventually consider only elite policies that form a lower-dimensional manifold of the parameter space.This drastically reduces the variability of solutions with similar BDs, and empirically highly improves the transfer of proximity from BDs to the trajectory level. Whether we can rely more on this property to extend our results will be part of our future work.
>
> [1] Discovering the Elite Hypervolume by Leveraging Interspecies Correlation, Vassiliades et al. 2018
> [2] Policy Manifold Search: Exploring the Manifold Hypothesis for Diversity-based Neuroevolution, Rakicevic et al. 2020
>
> - **I wonder if there is a better update schedule than random diversity and quality updates. The authors mention joint updates performed worse. It's not necessary, but I suggest the authors experiment with annealing/oscillating approaches to DPG/QPG ratios for the population, with a larger period than a single update. My main concern is that perhaps QPG ascent benefits from consecutive updates to find better solutions faster, and the current approach "destroys" the gains from the gradient, leading to sample inefficiency.**
>
> Concerning the schedule for quality and diversity, the reviewer is right and we think that this is an exciting direction for future research. In this study, we observed that updating randomly and uniformly for either quality or diversity works well and tends to balance naturally between both criteria during training thanks to the grid selection mechanism. In future work, we will investigate annealing/oscillating approaches. Another exciting direction we would like to investigate is to meta-learn the correct proportions. We don't think that our method can "destroy" the gains from quality policy gradients thanks to the MAP-Elite grid mechanism. Indeed, when we apply a diversity update to a policy, the policy is copied first, then updated and evaluated. If the new policy is neither novel nor performing better, it is dropped and the previous policy is kept. We will mention these future directions in the final version as long as space constraints make it possible.

---

### Official Review · Reviewer_QfRA · 2021-07-16

**Rating:** 7
**Confidence:** 2

**Summary:**

The paper proposes a Diversity Policy Gradient for Quality-Diversity optimization method. The approach solves a problem of finding a collection of well performing policies in continuous control environments. Rather than searching for a single optimal solution, the method aims to find a diverse set of high-performing solutions. As a result, the paper solves exploration and control problems with improved sample-efficiency over its evolutionary competitors.


**Limitations And Societal Impact:**

Limitations are very well discussed.

**Main Review:**

The paper is well-written, clear, and easy to follow. It solves an interesting problem with improved sample-efficiency over other state-of-the-art algorithms. The authors considered a good set of experiments. The results and limitations are nicely discussed.

Minor comment: you may want to rewrite lines 170 and 171 in the Related Work section.


**Time Spent Reviewing:**

5

---

> ### Author Response · Authors · 2021-08-10
> **Response to Reviewer QfRA**
>
> Thank you for your work and your comments. We will rewrite lines 170 and 171 in the related work section.

---

### Author Response · Authors · 2021-08-10
**Main Response**

We sincerely thank all reviewers for their very useful feedback.

We apologize for lines 170 and 171, which are indeed forgotten comments when writing the paper that we will remove.

Additional comments specific to each reviewer are answered in specific answers. We will perform all the corresponding modifications in the final version of the paper upon acceptance.

---

### Decision · Program_Chairs · 2021-09-27

**Decision:**

Reject

**Comment:**

This paper proposes a novel method to learn a set of diverse and high-quality policies. Ablation studies are sufficient to verify that the results are the consequence of the algorithm components. All reviewers gave positive scores.

However, this is not the first work on learning diverse policies. For examples, learning diverse policies were commonly demanded in game AI [Generating Behavior-Diverse Game AIs with Evolutionary Multi-Objective Deep Reinforcement Learning. IJCAI 2020], and skill discovery also requires diverse sub-policies [Learning Diversity is All You Need: Learning Skills without a Reward Function. ICLR 2019]. The major weakness is that the previous studies on learning diverse policies/skill are all ignored.

Another weakness is that there is no theoretical guarantee on the convergence of the algorithm and on the level of diversity we can expect. This does not mean that a theoretical guarantee is a must. However, the algorithm design is heuristic. The outcome of the algorithm might be undesired. A discussion on the rationality of the algorithm on how much the diversity can be is expected.

Overall I think this paper contains novelty and contributions, but it should have done a complete survey of related work, contacted experiments comparing other diversity encourage methods, and discussed in-depth on the convergence property.